# Baseline Sensitivity and Resistance Detection of *Stemphylium lycopersici* to Pydiflumetofen

**DOI:** 10.3390/jof11100734

**Published:** 2025-10-11

**Authors:** Xiangyu Liu, Kexin Yang, Jie Wu, Qiuyan Bi, Fen Lu, Jiqiang Wang, Jianjiang Zhao

**Affiliations:** 1Plant Protection Institute, Hebei Academy of Agricultural and Forestry Sciences, Key Laboratory of Integrated Pest Management on Crops in Northern Region of North China, Ministry of Agriculture and Rural Affairs, P. R. China, IPM Innovation Center of Hebei Province, International Science and Technology Joint Research Center on IPM of Hebei Province, Baoding 071000, China; xiangyuliu@haafs.org (X.L.); yangkexin1130069@126.com (K.Y.); wujie@haafs.org (J.W.); 0304biqiuyan@haafs.org (Q.B.); 1lufen1206@haafs.org (F.L.); 2Syngenta (Shanghai) Crop Protection Technology Co., Ltd., Shanghai 200126, China; jiqiang.wang@syngentagroup.cn

**Keywords:** tomato gray leaf spot, *S. lycopersici*, pydiflumetofen, baseline sensitivity, resistance

## Abstract

Tomato gray leaf spot (TGLS), caused by *Stemphylium* spp., is a common disease leading to significant economic losses in tomato production. Pydiflumetofen is a novel succinate dehydrogenase inhibitor (SDHI) fungicide that has been registered for TGLS management. To evaluate the susceptibility of *S. lycopersici* to pydiflumetofen in tomato-producing regions of Hebei Province, we determined the sensitivity of 212 *S. lycopersici* isolates using mycelial growth inhibition. The sensitivity distribution exhibited a multimodal pattern. Resistance to pydiflumetofen was observed in some field isolates, with highly resistant isolates being identified in Chengde, Hengshui, and Tangshan. After removing outliers, the baseline sensitivity of *S. lycopersici* to pydiflumetofen was established, with a mean EC_50_ value of 1.0400 ± 0.0515 μg/mL. Sequence analysis revealed point mutations only in SdhC (SdhC^S73P^, SdhC^G79R^, SdhC^H134R^, SdhC^S135R^) among the resistant isolates. No significant differences were observed between certain resistant isolates (FQSL1-10 and FQSL1-14) and the sensitive isolates in temperature adaptability, mycelial growth rate, or pathogenicity. These results suggest that pydiflumetofen has high activity against TGLS, but integrated fungicide application is necessary for delaying resistance evolution in TGLS management.

## 1. Introduction

Tomato gray leaf spot (TGLS), caused by *Stemphylium* spp., occurs worldwide, especially in warm, moist areas [1]. *Stemphylium* spp. can infect multiple hosts, such as sweet potato, garlic, and eggplant in China [2,3,4]. *S. lycopersici*, *S. solani,* and *S. floridanum* are the major pathogens causing TGLS globally [5]. Based on ITS and GPD sequence analysis, *S. lycopersici* has been identified as the primary causal agent of TGLS outbreaks in China [6]. The pathogens primarily infect tomato leaves. In the initial stage, leaves develop small brownish spots that become slightly sunken. As the infection progresses, affected leaves turn yellow, become necrotic, and drop prematurely, reducing yield [7]. Symptoms can be observed on leaves at all stages of tomato growth, notably after wet weather or sprinkler irrigation. TGLS was first reported in Yuntai County, Shandong Province, China. Because of expansion of facility cultivation and frequent changes in tomato varieties, TGLS has caused substantial yield losses in China [8].

Chemical control remains the most effective strategy for managing TGLS. Fungicides with different modes of action have been registered to prevent the development and spread of TGLS. However, limited research exists on fungicide screening for TGLS control. Li et al. found that pyraclostrobin exhibited high efficacy against TGLS in vivo [9]. Diethofencarb-benomyl (45%, WP), pydiflumetofen-difenoconazole (200 g/L, SC), and pydiflumetofen-fludioxonil (400 g/L, SC) are registered for controlling TGLS in China (http://w.icama.cn/zwb/dataCenter, 2 August 2025). Succinate dehydrogenase inhibitors (SDHIs; FRAC group 7) target the ubiquinone-binding pocket (Q site) of succinate dehydrogenase (SDH) in the mitochondrial respiratory chain, inhibiting electron transfer, disrupting ATP production, and consequently suppressing fungal growth [10,11,12,13]. With advances in compound research, more efficient and broad-spectrum SDHI fungicides have been developed and applied [14]. To date, over 20 SDHI fungicides have been applied to control fungal pathogens. Pydiflumetofen (3-(difluoromethyl)-N-methoxy-1-methyl-N-(1-(2,4,6-trichlorophenyl) propan-2-yl)-1H-pyrazole-4-carboxamide) is a new fungicide developed by Syngenta and has been registered in Argentina, USA, the United Kingdom, Canada, and China [15]. It is applied in economic crops such as grapes, peanuts, melons, and peppers [16]. As a broad-spectrum fungicide, pydiflumetofen can prevent powdery mildew, leaf spot, brown spot, gray mold, fusarium head blight, sclerotinia rot disease, etc. [17,18]. It also exhibits nematode control activity and improves crop health [19]. Pydiflumetofen suppresses microorganisms by disrupting mitochondrial respiration through specific inhibition of succinate dehydrogenase (SDH; Complex II). Although SDHI fungicides demonstrate high efficacy against fungi, recent studies indicate that pathogen resistance to SDHIs is emerging [20]. Currently, FRAC classifies the resistance risk of SDHI fungicides as moderate to high [21]. Some pathogens have developed resistance to SDHI fungicides, such as *Fusarium graminearum*, *Alternaria alternata*, *Monilinia fructicola,* and *Botrytis cinerea* [22,23,24]. Their resistance mechanism involves amino acid mutations in SDH subunits. The primary resistance mechanism of pathogens involves point mutations in SDH subunits (SdhB, SdhC, SdhD) that reduce the pydiflumetofen-binding affinity. Thus, monitoring resistance development during SDHI application is essential.

In this study, the baseline sensitivity of *S. lycopersici* to pydiflumetofen was established, and the current resistance status of *S. lycopersici* to pydiflumetofen was determined in Hebei Province. The risk of resistance development was assessed by evaluating the biological fitness of resistant mutants. The results provide a basis for the effective application of pydiflumetofen in TGLS control.

## 2. Materials and Methods

### 2.1. Isolates and Culture Conditions

Diseased leaves were collected from tomato greenhouses in Hebei Province from 2021 to 2023. Lesion sites were excised with sterilized scalpels, surface-disinfected in 1% sodium hypochlorite for 3 min, rinsed with sterile water, treated with 70% ethanol for 3 min, washed three times with sterile water, and transferred to PDA plates.

Upon growth of hyphal tips from the tissue until hyphae appeared on the PDA plates in an incubator at 25 °C, the suspected *S. lycopersici* isolates were selected and transferred to new PDA plates. All isolates were purified by the hyphal tip separation isolation method and stored at −20 °C for further analysis. These isolates were characterized morphologically and identified molecularly using and internal transcribed spacer (ITS1: 5′-TCCGTAGGTGAACCTGCGG-3′ and ITS4: 5′-TCCTCCGCTTATTGATATGC-3′).

### 2.2. Fungicides and Reagents

Fungicide information: Pydiflumetofen (98% a.i., Syngenta Crop Protection Technology Co., Ltd., Shanghai, China) was dissolved in acetone to prepare a 10^4^ μg/mL stock solution.

Potato dextrose agar (PDA, 200 g potato, 20 g dextrose, and 20 g agar per liter of deionized water) was used for mycelial growth and sensitivity tests for pydiflumetofen.

### 2.3. Establishment of Baseline Sensitivity and Determination of Resistance Level

The sensitivity of *S*. *lycopersici* to pydiflumetofen was determined using the mycelial growth inhibition method. The assays were conducted on PDA amended with pydiflumetofen at concentrations of 0, 0.005, 0.01, 0.05, 0.1, 0.5, 1, 2, 5, 10, 20, 40, and 50 μg/mL, and acetone-amended PDA was used as a control. After 5 days incubation in dark at 25 °C, the colony diameters were assessed using the crossover method. The growth-inhibitory rate was calculated as follows: Inhibition rate (%) = (control diameter − treatment diameter)/(control diameter − 5) × 100. The EC_50_ value for each isolate was calculated according to a previous study [25]. The experiment was conducted twice, with three replications each time. ANOVA was carried out according to SPSS24.0. The same letter after the number in the same column are not significantly different according to the least significant difference (LSD) test at *p* = 0.05. The EC_50_ values of 108 isolates following a normal distribution were used to calculate the mean EC_50_ for evaluating the resistance level of *S. lycoperisici* isolates. Resistance levels were classified by Resistance Factor (RF). RF = EC_50_ value of isolate/mean EC_50_. Isolates with an RF of less than 10 were classified as sensitive; those with an RF of 10–50 were classified as low-resistance; isolates with an RF between 50 and 100 were classified as medium-resistance; and isolates with an RF greater than 100 were classified as high-resistance.

### 2.4. Amplification and Sequence Analysis of Sdhs Genes from S. lycopersici

To clarify the mutation types of *S. lycopersici* isolates in response to pydiflumetofen, the four *Sdh* genes of 15 randomly selected isolates (1 sensitive, 8 low-resistance, 3 medium-resistance, and 3 high-resistance) were sequenced. Primers for amplifying the *SdhA*, *SdhB*, *SdhC*, and *SdhD* genes are listed in Table 1. PCR amplifications were performed in 25 μL of reaction mixture containing 50 ng of genomic DNA, 12.5 μL 2 × San Taq PCR Master Mix (Sangon Biotech Co. Ltd., Shanghai, China), and 1 μL of each primer (10 mmol L^–1^). PCR was performed in a Biometra TOne 96 G (Analytik Jena, Jena, Germany) with the following program: 95 °C for 3 min, followed by 34 cycles of denaturation at 95 °C for 30 s, annealing at 57 °C for 30 s, and extension at 72 °C for 1 min, with a final extension at 72 °C for 5 min. Sangon Biotech Co. Ltd. (Shanghai, China) sequenced the products. Bioedit v7.0.9 was used to predict and compare the amino acid sequence of the Sdhs to those of resistant isolates and the sensitive isolate.

### 2.5. Biological Fitness Assay for Different Sensitive Types of S. lycopersici to Pydiflumetofen

For the mycelial growth assay, 5 mm mycelial plugs from 5-day-old colonies were cultured on PDA at 15, 25, and 35 °C for 5 days in darkness. Colony diameters were measured, and the differences in mycelial growth between different sensitive type isolates were analyzed. Pathogenicity analysis was carried out according to a previously described method [26]. Tomato leaves were cut from healthy plants and punctured on one side with a sterilized needle. Mycelial plugs (5 mm in diameter) were cut from a 5-day-old colony edge and inoculated at the punctured site. Then leaves were cultured in a moist chamber at 25 °C for 5 days under dark conditions. The lesion diameter of each was measured by the crossover method. Six replicate leaves were used for each isolate, and the experiments were performed three times.

## 3. Results

### 3.1. Sensitive Baseline of S. lycopersici to Pydiflumetofen

The sensitivity of 212 isolates to pydiflumetofen was measured by mycelial growth inhibition. The sensitivity distribution of *S. lycopersici* to pydiflumetofen was plotted based on the frequency of EC_50_ (Figure 1). The frequency distribution of EC_50_ values for pydiflumetofen was multimodal, demonstrating the existence of resistant subpopulations in the field. To establish standard criteria for classifying the resistance levels of *S. lycopersici* populations to pydiflumetofen, the first subpopulation peak in distribution was established as the baseline sensitivity for pydiflumetofen. After removing outliers, 108 isolates were used to determine the baseline sensitivity (Figure 2). The EC_50_ value ranged from 0.0318 to 2.3606 μg/mL, with a mean EC_50_ of 1.0400 ± 0.0515 μg/mL. The distribution had a kurtosis of −0.294 and skewness of 0.233. A Kolmogorov–Smirnov test confirmed that this distribution could serve as the baseline sensitivity for *S. lycopersici* to pydiflumetofen in Hebei Province.

### 3.2. Sensitivity of S. lycopersici to Pydiflumetofen in Hebei Province

To further characterize the sensitivity of *S. lycopersici* populations, the isolates were classified into different resistance levels according to the resistance multipliers in the collection areas (Table 2). The results revealed predominant sensitivity to pydiflumetofen, but resistant isolates were detected in Hebei Province. The geographic distribution of highly resistant isolates varied among regions, with Chengde, Hengshui, and Tangshan harboring three, five, and two isolates, respectively. Medium-resistant isolates were detected in Hengshui (one), Shijiazhuang (one), and Tangshan (one). Chengde had the highest proportion of resistant isolates (58.33%), whereas Handan had the lowest (5.43%), despite having the largest sample size.

### 3.3. Mutation Analysis of Sdh Amino Acid of S. lycopersici

To analyze the resistance mechanism of *S. lycopersici* to pydiflumetofen, *Sdh* genes of 14 randomly selected isolates (1 sensitive, 9 low-resistance, 3 medium-resistance, and 3 high-resistance) were sequenced, and their amino acid sequences were compared. No mutations were found in *SdhA*, *SdhB,* and *SdhD*, but mutations were detected in *SdhC*. A mutation at position G79R was identified in FQSL1-10 and FQSL1-14. A mutation at position S135R was identified in DXSL1-8. A mutation at position H134R was identified in XTSL1-14. A mutation at position S73P was identified in 10 resistant isolates (Table 3).

### 3.4. Biological Fitness of Pydiflumetofen-Resistant Isolates of S. lycopersici

Resistant and sensitive isolates were compared for mycelial growth at 15 °C, 25 °C, and 35 °C, as well as for pathogenicity (Table 4). Compared with the sensitive isolate HSSL2-7, mycelial growth of FQSL1-10, DXSL1-6, and RYSL1-7 was decreased, but others did not differ at 15 °C. At 35 °C, the resistant isolates FQSL1-14 and DXSL1-6 exhibited a decline in growth rate relative to HSSL2-7. All isolates showed no difference in mycelial growth rate at 25 °C. Pathogenicity analysis revealed that resistant isolates exhibited smaller lesion sizes than sensitive isolates, except for FQSL1-10 and FQSL1-14. These results showed that the biological fitness of the resistant isolates FQSL1-10 and FQSL1-14 was similar to that of the sensitive isolate (HSSL2-7).

## 4. Discussion

Pydiflumetofen inhibits microbial energy production by disrupting the tricarboxylic acid cycle [27] and has shown high efficacy against *Botrytis cinerea*, *Alternaria solani*, *Zymoseptoria tritici*, and *Monilinia fructicola* [26]. Establishing the baseline sensitivity of *S. lycoperisici* to pydiflumetofen is critical for TGLS management. In this study, the sensitivity of 212 *S. lycopersici* isolates to pydiflumetofen exhibited a multimodal distribution. After removing outliers, the baseline sensitivity of *S. lycoperisici* to pydiflumetofen was established. The EC_50_ values for pydiflumetofen ranged from 0.0318 to 2.3606 μg/mL, with a mean value of 1.0400 ± 0.0515 μg/mL. Although resistant isolates were detected, sensitive isolates still accounted for the majority of the population in Hebei Province. These results provide a reference for monitoring the susceptibility of *S. lycopersici* to pydiflumetofen in field populations.

Within this research, pydiflumetofen-resistant isolates of *S. lycopersici* were first detected in the field populations. The sensitivity to pydiflumetofen showed significant regional variation across Hebei Province. Whereas no moderately or highly resistant isolates were detected in Handan and Baoding, highly resistant isolates were detected in Chengde, Hengshui, and Shijiazhuang. These regional differences may be linked to variations in environmental conditions and fungicide application. High-frequency application of SDHI fungicides drives selection pressure on pathogen populations. Therefore, continuous monitoring of *S. lycopersici*’s susceptibility to pydiflumetofen in tomato-producing regions is recommended.

Previous studies on pydiflumetofen’s resistance mechanisms have primarily utilized laboratory-induced mutants. For example, mutations ChSdhB^H277Y^, ChSdhB^I279T^, and ChSdhD^H133Y^ in *Colletotrichum heterostrophus* showed high levels of resistance to pydiflumetofen [28]. The resistance to pydiflumetofen in *Fusarium graminearum* is associated with mutations in the SdhB, SdhC, and SdhD [29]. But in this study, only SdhC^S73P^, SdhC^G79R^, SdhC^H134R^, and SdhC^S135R^ were detected in the resistant isolates, with SdhA, SdhB, and SdhD remaining conserved.

Biological fitness assays revealed that resistant isolates (FQSL1-10 and FQSL1-14) and the sensitive isolate (HSSL2-7) had no significant differences in mycelial growth (25 °C) and pathogenicity. This implied that there is a potential risk of pydiflumetofen-resistant isolates becoming dominant in the field. As single-site fungicides, rapid adoption of SDHI fungicides has accelerated resistance development. While pydiflumetofen demonstrates high efficacy against TGLS, long-term exclusive use accelerates resistance evolution. Thus, the combined use of fungicides with different mechanisms of action is crucial in TGLS management and to delay resistance evolution.

## 5. Conclusions

This study established the baseline sensitivity of *S. lycopersici* to pydiflumetofen and mapped the resistance distribution in Hebei Province. Mutations in resistant isolates were identified in SdhC, including SdhC ^S73P^, SdhC ^G79R^, SdhC ^H134R^, and SdhC ^S135R^. The biological fitness of some resistant isolates was similar to that of sensitive isolates. To mitigate resistance risk, pydiflumetofen should be alternated with fungicides with other modes of action in TGLS management.

## Figures and Tables

**Figure 1 jof-11-00734-f001:**
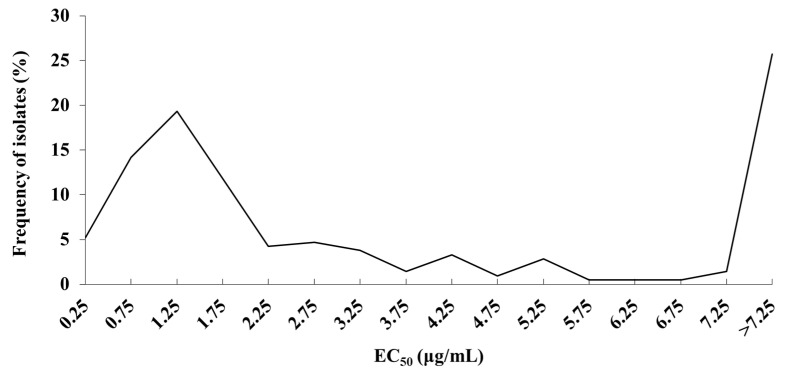
Frequency distribution of sensitivity to pydiflumetofen in 212 *S. lycopersici* isolates from Hebei Province.

**Figure 2 jof-11-00734-f002:**
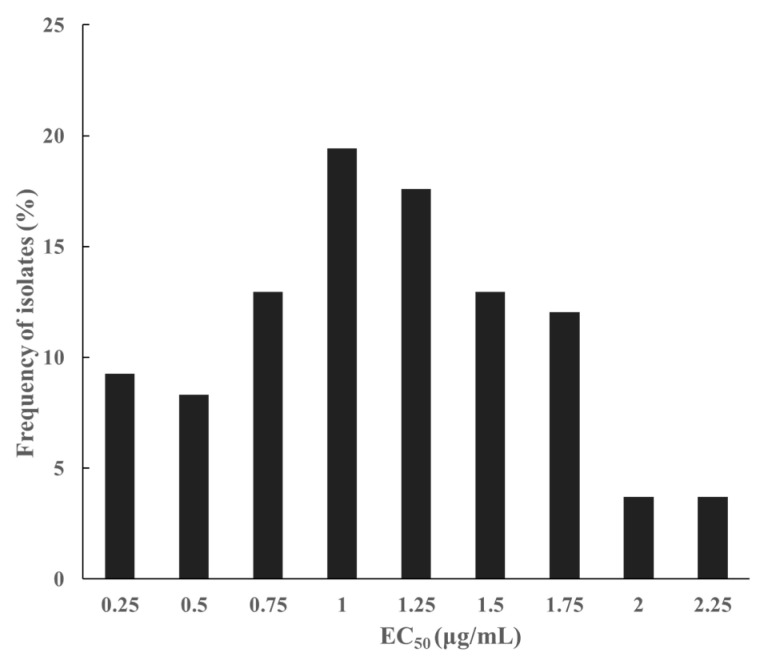
Frequency distribution of pydiflumetofen EC_50_ values in 108 *S. lycopersici* isolates.

**Table 1 jof-11-00734-t001:** Primers used for amplification of *Sdh*s genes of *S. lycopersici*.

Primer (5′→3′)	Sequence
*SdhA*-F	AAAAGTCGGGATGGCAGGTA
*SdhA*-R	TGGACCATGTGTGATCGTCT
*SdhB*-F	TGTCATAACCGAGGGAAGCT
*SdhB*-R	TCGTTGAGTGGGTGGATGAT
*SdhC*-F	ATCCACCCTCTCGACGATTC
*SdhC*-R	TTGACCTTCTCGCCATCCTT
*SdhD*-F	AAGACGACTGATACTGGGGC
*SdhD*-R	TGGCCGTACTGAGGTTGATT

**Table 2 jof-11-00734-t002:** Distribution of pydiflumetofen resistance levels in *S. lycopersici* populations (LR: low resistance; MR: medium resistance; HR: high resistance).

-	Number of Isolates	Isolate/Percentage of the Total Population (%)			
		Sensitive	LR	MR	HR
Baoding	32	30 (93.75)	2 (6.25)	0 (0.00)	0 (0.00)
Chengde	12	5 (41.67)	4 (33.33)	0 (0.00)	3 (25.00)
Handan	92	87 (94.57)	5 (5.43)	0 (0.00)	0 (0.00)
Hengshui	27	14 (51.85)	7 (25.93)	1 (3.70)	5 (18.52)
Shijiazhuang	24	14 (58.33)	9 (37.50)	1 (4.17)	0 (0.00)
Tangshan	25	16 (64.00)	6 (24.00)	1(4.17)	2 (8.33)
Total	212	166 (78.30)	33 (15.57)	3 (1.42)	10 (4.72)

**Table 3 jof-11-00734-t003:** Point mutation types of four Sdh amino acid sequences in 14 *S. lycopersici* isolates with different resistance levels to pydiflumetofen (LR: low resistance; MR: medium resistance; HR: high resistance).

Name of Isolates	Region	Phenotype	Mutation Type			
			SdhA	SdhB	SdhC	SdhD
HSSL2-7	Baoding	S	-	-	-	-
FQSL1-10	Shijiazhuang	LR	-	-	G79R	-
FQSL1-14	Shijiazhuang	LR	-	-	G79R	-
DXSL1-8	Handan	LR	-	-	S135R	-
XTSL1-14	Handan	LR	-	-	H134R	-
CYSL1-7	Tangshan	HR	-	-	S73P	-
DXSL1-6	Baoding	LR	-	-	S73P	-
DXSL1-10	Hengshui	MR	-	-	S73P	-
DXSL2-8	Shijiazhuang	MR	-	-	S73P	-
DXSL3-5	Handan	LR	-	-	S73P	-
FQSL1-6	Tangshan	LR	-	-	S73P	-
RYSL1-7	Hengshui	HR	-	-	S73P	-
RYSL1-9	Hengshui	LR	-	-	S73P	-
TSSL2-12	Hengshui	HR	-	-	S73P	-
TSSL2-27	Hengshui	LR	-	-	S73P	-

**Table 4 jof-11-00734-t004:** Biological fitness of pydiflumetofen-sensitive and -resistant *S. lycopersici* isolates.

Isolates	Phenotype	Mycelial Growth Rate (mm/d)			Lesion Size (mm)
		15 °C	25 °C	35 °C	
HSSL2-7	S	3.48 ± 0.16 ab	8.07 ± 0.26 ab	5.62 ± 0.08 ab	15.85 ± 1.07 a
FQSL1-10	LR	2.86 ± 0.08 c	7.86 ± 0.29 b	5.40 ± 0.15 bc	17.98 ± 0.76 a
FQSL1-14	LR	3.14 ± 0.25 bc	7.64 ± 0.19 b	5.33 ± 0.08 c	16.73 ± 0.52 a
DXSL1-8	LR	3.67 ± 0.26 a	8.48 ± 0.16 a	5.52 ± 0.08 abc	14.97 ± 0.50 b
XTSL1-14	LR	3.60 ± 0.04 a	8.10 ± 0.41 ab	5.55 ± 0.21 abc	14.72 ± 0.56 b
CYSL1-7	HR	3.76 ± 0.22 a	8.48 ± 0.22 a	5.70 ± 0.04 a	13.33 ± 1.71 b
DXSL1-6	LR	2.88 ± 0.04 c	8.07 ± 0.07 ab	5.38 ± 0.08 c	14.47 ± 0.63 b
RYSL1-7	HR	2.81 ± 0.08 c	8.05 ± 0.16 ab	5.40 ± 0.10 bc	13.43 ± 0.70 b
TSSL2-12	HR	3.67 ± 0.36 a	8.02 ± 0.18 ab	5.55 ± 0.15 abc	13.67 ± 0.44 b

ANOVA analysis was carried out SPSS24.0. The same letter after the number in the same column are not significantly different according to the least significant difference (LSD) test at *p* = 0.05.

## Data Availability

The original contributions presented in this study are included in the article. Further inquiries can be directed to the corresponding author.

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
