# Peer review of "Baseline Sensitivity and Resistance Detection of Stemphylium lycopersici to Pydiflumetofen"

_jof, 2025, doi:10.3390/jof11100734_

Round 1

Reviewer 1 Report

Comments and Suggestions for Authors

This study investigates the effect of pydiflumetofen (Pyd) on the isolates of Stemphylium lycopersici. Pyd resistance can result from specific mutations in the genes that code for the succinate dehydrogenase (Sdh) enzyme complex, specifically the SdhB and SdhC subunits. The presented study also included the sequence analysis of Sdhs genes (A, B, C and D) of 15 S. lycopersici isolates.

This topic is of interest and there is some research on the different fungal species developing resistance to Pyd. This study is a nice addition to the previous studies.

Introduction is proper for the paper, but the mode of pydiflumetofen action should be added. This is noted in the discussion, but mention it here in short.

The methodology is fine, written in detail. I do not have major remarks, but minor comment:

Line 95: Add remark for the acetone that itself may have antifungal activity and can affect fungal growth. Consider this.

The results section is nicely divided in subchapters presenting obtained results.

Remarks:

Line 157: Do not start a sentence with a number

Line 176: Delete showed

The discussion section is rather short compared with a few literature data. This part has many grammar errors, as: Line 209: of 202; Line 217: were instead of are; Line 232: had instead of are;...

The conclusion is consistent with presented results. Maybe to add the classification of the resistance of S. lycopersici to Pyd based on your research. (i.e. We concluded that the resistance of S. lycopersici to Pyd was classified as low/moderate/high risk.) 

Cited references are adequate.

Figures are illustrative and well presented

Some corrections:

Figure 1: Y-axis Frequency of isolates

Figure 2: Y-axis Frequency of isolates

Comments on the Quality of English Language

I am not a native English speaker, but the manuscript should be proofread

Author Response

Comments1: Introduction is proper for the paper, but the mode of pydiflumetofen action should be added. This is noted in the discussion, but mention it here in short.

Response1:Thank you for pointing this out. We agree with this comment. Therefore, we briefly describe the mode of pydiflumetofen action. “Pydiflumetofen could suppress microorganisms by disrupting mitochondrial respiration through specific inhibition of succinate dehydrogenase (SDH; Complex II).”, “The primary resistance mechanism of pathogens involves point mutations in SDH subu-nits (SdhB, SdhC, SdhD) that reduce the pydiflumetofen 's binding affinity.”

Comments2: The methodology is fine, written in detail. I do not have major remarks, but minor comment:

Line 95: Add remark for the acetone that itself may have antifungal activity and can affect fungal growth. Consider this.

Response2:Thank you for pointing this out. We agree with this comment. We took this into consideration during the experimental process, so we add acetone-amended PDA in control and annotated it in the methods.  

Comments3: The results section is nicely divided in subchapters presenting obtained results.

Remarks:

Line 157: Do not start a sentence with a number

Line 176: Delete showed

Response3: Thank you for pointing this out. We agree with this comment.

“The geographic distribution of highly resistant isolates varied among regions, with Chengde, Hengshui, and Tangshan harboring 3, 5, and 2 isolates, respectively.” were used in Line157. “showed” were deleted in Line176.

The discussion section is rather short compared with a few literature data. This part has many grammar errors, as: Line 209: of 202; Line 217: were instead of are; Line 232: had instead of are;...The conclusion is consistent with presented results. Maybe to add the classification of the resistance of S. lycopersici to Pyd based on your research. (i.e. We concluded that the resistance of S. lycopersici to Pyd was classified as low/moderate/high risk.) 

Cited references are adequate.

Figures are illustrative and well presented

Some corrections:

Figure 1: Y-axis Frequency of isolates

Figure 2: Y-axis Frequency of isolates

Response4: Thank you for pointing this out. We agree with this comment. We replaced “202” with “212” in Line 209; “were” instead of “are” in Line 217; “had” instead of “are: in Line232. Figuer1 and Figuer2 have been corrected.

Rievewer2

Comments1: L89-90, p3: please provide more detailed information about the procedures of the hyphal tip separation isolation method. 

Response1: Thank you for pointing this out. We agree with this comment. We added some details about hyphal tip separation isolation method in method. “Upon growth of hyphal tips from the tissue until hyphae appeared on the PDA plates in an incubator at 25°C,…”

Reviewer 2 Report

Comments and Suggestions for Authors
  • This paper describes analysis of the sensitivity of Stemphylium lycopersici, which causes tomato gray leaf spot (TGLS), to pydiflumetofen, a fungicide used for management of TGLS disease. The authors investigated the sensitivity of S. lycopersici isolates collected from diseased tomato leaves grown in greenhouses and showed that there were diverse degrees of sensitivity ranging from sensitive to high resistance. They also found that point mutations were detected in a specific gene encoding succinate dehydrogenase, a target enzyme for this fungicide. These findings are novel and important for practical application of pydiflumetofen in Agriculture. I have only minor concerns as described below.

    There are many grammatical and typographical errors throughout the manuscripts. The manuscript should be edited by native English-speaking researchers.

  • comments_authors_detail:

    L89-90, p3: please provide more detailed information about the procedures of the hyphal tip separation isolation method. 

    L202,p7: The description on statistics should be moved to the materials and methods section.

    Figures 1 and 2: overall, this is a lack of caption in each figure. 

    L18, p1, L148-149, p4, and L212-213, p8: it seems that the number of the signicicant digits for EC50 of pydiflumetofen is too many. Appropriate number of signicicant digits should be used.

Author Response

Comments1: L89-90, p3: please provide more detailed information about the procedures of the hyphal tip separation isolation method. 

Response1: Thank you for pointing this out. We agree with this comment. We added some details about hyphal tip separation isolation method in method. “Upon growth of hyphal tips from the tissue until hyphae appeared on the PDA plates in an incubator at 25°C,…”

Comments2: L202 p7: The description on statistics should be moved to the materials and methods section.

Response2: Thank you for pointing this out. We agree with this comment. We moved statistics analysis to the materials and methods section.

Comments3: Figures 1 and 2: overall, this is a lack of caption in each figure. 

L18, p1, L148-149, p4, and L212-213, p8: it seems that the number of the signicicant digits for EC50 of pydiflumetofen is too many. Appropriate number of signicicant digits should be used.

Response3: Thank you for pointing this out. We agree with this comment. We used signicicant digits in this description.

Author Response

Thank you for pointing this out. We agree with these comment. 

Round 2

Reviewer 2 Report

Comments and Suggestions for Authors

The manuscript  has been improved.

Comments on the Quality of English Language

There still remain many typographical and grammatical errors. The authors should recheck the manuscript carefully. 

Author Response

Comments1:There still remain many typographical and grammatical errors. The authors should recheck the manuscript carefully. 

Response1:Thank you for pointing this out. We agree with this comment. Therefore, we  have made a lot of modifications. Please refer to the manuscript for details